# Machine Learning Models for Prediction of Severe *Pneumocystis carinii* Pneumonia after Kidney Transplantation: A Single-Center Retrospective Study

**DOI:** 10.3390/diagnostics13172735

**Published:** 2023-08-23

**Authors:** Yiting Liu, Tao Qiu, Haochong Hu, Chenyang Kong, Yalong Zhang, Tianyu Wang, Jiangqiao Zhou, Jilin Zou

**Affiliations:** 1Department of Organ Transplantation, Renmin Hospital of Wuhan University, Wuhan 430060, China; 2Department of Urology, Renmin Hospital of Wuhan University, Wuhan 430060, China; 3Department of Nephrology, Renmin Hospital of Wuhan University, Wuhan 430060, China

**Keywords:** *Pneumocystis carinii* pneumonia (PCP), machine learning models, predict, artificial intelligence

## Abstract

Background: The objective of this study was to formulate and validate a prognostic model for postoperative severe *Pneumocystis carinii* pneumonia (SPCP) in kidney transplant recipients utilizing machine learning algorithms, and to compare the performance of various models. Methods: Clinical manifestations and laboratory test results upon admission were gathered as variables for 88 patients who experienced PCP following kidney transplantation. The most discriminative variables were identified, and subsequently, Support Vector Machine (SVM), Logistic Regression (LR), Random Forest (RF), K-Nearest Neighbor (KNN), Light Gradient Boosting Machine (LGBM), and eXtreme Gradient Boosting (XGB) models were constructed. Finally, the models’ predictive capabilities were assessed through ROC curves, sensitivity, specificity, accuracy, positive predictive value (PPV), negative predictive value (NPV), and F1-scores. The Shapley additive explanations (SHAP) algorithm was employed to elucidate the contributions of the most effective model’s variables. Results: Through lasso regression, five features—hemoglobin (Hb), Procalcitonin (PCT), C-reactive protein (CRP), progressive dyspnea, and Albumin (ALB)—were identified, and six machine learning models were developed using these variables after evaluating their correlation and multicollinearity. In the validation cohort, the RF model demonstrated the highest AUC (0.920 (0.810–1.000), F1-Score (0.8), accuracy (0.885), sensitivity (0.818), PPV (0.667), and NPV (0.913) among the six models, while the XGB and KNN models exhibited the highest specificity (0.909) among the six models. Notably, CRP exerted a significant influence on the models, as revealed by SHAP and feature importance rankings. Conclusions: Machine learning algorithms offer a viable approach for constructing prognostic models to predict the development of severe disease following PCP in kidney transplant recipients, with potential practical applications.

## 1. Background

Due to the prolonged use of immunosuppressive medications, the immune response of kidney transplant recipients is compromised, making them susceptible to severe complications such as *Pneumocystis carinii* pneumonia (PCP). PCP progresses rapidly and can lead to irreversible organ damage and even death. The risk of PCP is high up to 1 year after kidney transplantation, especially up to 6 months after surgery, and in addition, according to studies, PCP can occur after 1 year after surgery even if the kidney transplant recipient has received prophylactic treatment for 6–12 months [1,2]. The overall incidence of PCP after kidney transplantation ranges from 0.3% to 2.6%, with a mortality rate of up to 50% [3]. Therefore, early identification of patients at risk of developing severe *Pneumocystis carinii* pneumonia (SPCP) during hospitalization is crucial for timely intervention and improved clinical management [4].

Given the complexity of PCP in kidney transplant recipients and the numerous factors influencing disease progression, relying solely on traditional statistical methods for clinical assessment may not accurately predict the likelihood of SPCP. However, machine learning algorithms offer a promising approach by integrating multiple clinical and laboratory variables to enhance predictive accuracy, albeit at the cost of interpretability. Currently, there is limited research on PCP prediction. Previous studies by Tang et al. [5] and Elie et al. [6] have developed prediction models for PCP development in immunosuppressed HIV-negative patients and ICU patients with hematologic malignancies and acute respiratory failure, respectively. Cai et al. [7] developed a prediction model for PCP in CKD patients using decision trees and Nomogram plots, while Wan et al. [8] established a prediction model for invasive mechanical ventilation (IMV) in 148 PCP patients using Nomogram plots. However, to date, no machine learning-based prognostic model specifically targeting SPCP in kidney transplant recipients has been developed.

In this study, we enrolled 88 post-kidney transplant patients with PCP and utilized a machine learning approach to predict the risk of developing SPCP based on common laboratory test results and clinical presentations at admission. We analyzed the most important laboratory parameters that can be used for early identification of patients at a high risk of developing SPCP and increased mortality.

## 2. Methods

### 2.1. Study Population

The study population comprised patients who underwent kidney transplantation and were admitted to the Organ Transplantation Department of Renmin Hospital of Wuhan University between April 2018 and April 2022. Ultimately, a total of 88 patients with PCP were included in the study. The primary outcome of interest was the occurrence of SPCP during hospitalization. SPCP was defined as patients with hypoxaemia or respiratory failure requiring non-invasive ventilation with at least 50% FiO_2_ (partial oxygen inhalation) or invasive ventilation [9]. The diagnosis of SPCP was established by two physicians with advanced clinical expertise. Secondary outcome measures included patient mortality and laboratory test results at the time of discharge. Due to the limitations of bedside oxygen therapy in general hospital beds, all SPCP were admitted to the ICU for treatment in this study, and none of the general PCP patients were admitted to the ICU for treatment.

### 2.2. Variable Collection

We collected various patient information, including: (I) Demographic data such as age, gender, and time since kidney transplantation. (II) Clinical manifestations upon admission, such as fever, dry cough, and progressive dyspnea. (III) Laboratory results at admission, including glactomannan (GM) testing, C-reactive protein (CRP), procalcitonin (PCT), white blood cell count (WBC), lymphocyte count (LYM), neutrophil count (Neu), hemoglobin (Hb), albumin (ALB), globulin (GLB), platelet count (PLT), blood urea nitrogen (BUN), blood glucose (Glu), serum creatinine (sCr), CD4 count, and presence of cytomegalovirus (CMV) infection. (IV) Past medical history, including prior ATG induction and episodes of rejection. (V) Additionally, we monitored the patients’ outcome measures, including length of hospital stay, admission to the ICU, and mortality. We used the KNN method to interpolate missing data. By collecting these variables, we aimed to capture comprehensive information about the patients’ characteristics, clinical presentation, laboratory findings, medical history, and subsequent prognostic outcomes.

### 2.3. Data Preprocessing

We divided the overall dataset into training and test sets using a 7:3 ratio. Categorical variables were encoded using one-hot encoding, while continuous variables were standardized using the min–max scaling technique, which rescales the variable values to a range of 0–1, ensuring comparability across dimensions. To address the issue of imbalanced data, we employed the Synthetic Minority Over-sampling Technique (SMOTE) on the training set [10]. SMOTE offers a valuable solution by generating synthetic samples of the minority class, effectively improving its representation and enhancing the model’s ability to learn from the data. By oversampling the minority class, SMOTE helps mitigate bias in predictions, enhances accuracy, and reduces the risk of overfitting.

### 2.4. Variable Screening

In the SMOTE-treated training set, we employed LASSO regression to reduce the dimensionality of the data variables and identify the most relevant variables for predicting the probability of SPCP upon admission using the optimal lambda. Lasso regression provides effective variable filtering and prediction capabilities due to its advantages in feature selection and regularization. It efficiently selects relevant variables while suppressing noise, improving model interpretability, and reducing overfitting.

### 2.5. Correlation Analysis

Furthermore, we conducted Spearman correlation analysis to examine the interrelationships among the variables after LASSO regression screening. This allowed us to assess for potential multicollinearity, ensuring that there were no significant correlations among the variables. By evaluating multicollinearity, we ensured the robustness of our model and minimized any adverse effects on the interpretability and reliability of the results.

### 2.6. Hyperparameter Optimisation

Six machine learning methods, including LR, SVM, XGB, RF, LGBM, and KNN, were selected for hyperparameter optimization in the training set, and a 5-fold cross-validation approach was used. The 5-fold cross-validation involved dividing the training set into five subsets, four of which were used as the internal training set and one as the internal validation set. Exploration was carried out. The average performance was calculated and the hyperparameters were adjusted using a grid search to maximize the AUC of the internal validation set.

### 2.7. Model Evaluation

We fitted the machine learning models using the optimal hyperparameters and evaluated the models on a test set. We compared different models based on their AUC values, sensitivity, specificity, accuracy, NPV, PPV, and F1-score.

### 2.8. SHAP and Variable Importance of the Best Model

We utilized the SHAP framework to generate visualizations of the optimal model, showcasing the variable importance in predicting the probability of developing SPCP disease after hospitalization. SHAP, a robust methodology for interpreting machine learning models, leverages Shapley values derived from cooperative game theory to measure the impact of each feature on the model’s output. By applying SHAP, we are able to analyze the importance of individual variables.

### 2.9. Statistical Analysis

Statistical analysis of the data was performed using R language (V4.2.1) and Python (3.11.3). Continuous variables were assessed using independent t-tests or Mann–Whitney U-tests and reported as mean ± standard deviation or median with interquartile range. Categorical variables were expressed as numbers and percentages and analyzed using the Chi-square test or Fisher’s exact test.

## 3. Results

### 3.1. Patients and Variables

In this study, a total of 88 patients who underwent kidney transplantation and developed PCP were included. The inclusion criteria were as follows: 1. Radiological evidence of pneumonia with detection of *P. jirovecii* in sputum samples or peripheral blood using metagenomic next-generation sequencing (mNGS). 2. Clinical diagnosis was based on clinical characteristics, typical pulmonary CT imaging changes, elevated 1,3-β-D-glucan levels, and the sensitive response to anti-PCP therapy [11]. Patients who met the following exclusion criteria were not included in the study: (1) Patients who died upon admission. (2) Recipients of multi-organ transplantation. A flowchart of the process of this study is shown in Figure 1. All patients met the diagnostic criteria for PCP. Based on the occurrence of SPCP, the patients were divided into two groups: 18 patients in the SPCP group and 70 patients in the non-severe PCP (NSPCP) group. Among the included patients, 28 were female and 60 were male, as shown in Table 1. The dataset was divided into a training set and a test set using a 7:3 ratio. Statistical differences were observed between the training and test sets for gender, Scr at admission, and concurrent CMV infection (*p* < 0.05), while no statistical differences were observed for other variables (*p* > 0.05). Variables such as symptoms at admission, admission data, and medical history (excluding prognosis information such as ICU, death, and duration of hospitalization) were selected as potential predictors. Feature selection was carried out using a LASSO regression with 5-fold cross-validation (as shown in Figure 2), using the “minimum plus one standard error” criterion to identify the optimal penalization coefficient lambda (λ), and finally, five variables were identified. The correlation heat map showed no significant correlation between the five variables screened (as shown in Figure 3). The results of the analysis of covariance showed that the variance inflation factor (VIF) between each variable was less than 10, indicating the absence of multicollinearity (Appendix A).

### 3.2. Parameter Optimization and Model Fitting

Due to the imbalance of the dataset, which can lead to model instability, we applied the SMOTE technique to the training set. This resulted in a balanced training set consisting of 48 cases of NSPCP and 48 cases of SPCP. After standardizing the variables, we selected the five variables identified in the training set and used them as inputs for the machine learning classifiers to construct the prediction model. We adjusted the model parameters using the average optimal AUC value, and the specific parameter settings are provided in Appendix A. The five-fold cross-validated ROC curves of the internal validation set are shown in Figure 4. It can be observed that, except for the LGBM model, the average AUC values of the other five models in the internal validation set exceeded 90%. The KNN model performed the best in terms of AUC, PPV, and F1_Score. The LR model demonstrated the highest accuracy and NPV. The XGB and SVM models performed optimally in terms of sensitivity and specificity, respectively. The RF model performs well in terms of overall performance. Detailed modelling results are provided in Appendix A. Since the training set was primarily used for parameter optimization, we focused on comparing the performance of different models in the validation set. We found that the RF model outperformed the others in the validation set, except for specificity. Meanwhile, the XGB and GBM models demonstrated the best performance in terms of specificity. The detailed test set model results are shown in Table 2 and Figure 5 and Figure 6.

### 3.3. Model Interpretation

To help explain the impact of each feature on the model outcome, we employed the SHAP algorithm. Each row in the graph represents a feature, sorted from top to bottom based on feature importance. The *x*-axis represents the SHAP values, while the points indicate the samples, and the color represents the feature values (red indicating high and blue indicating low). We selected the RF model, which exhibited the best overall performance in the test set, to interpret the feature importance. Additionally, we presented the ranking of each variable’s importance for the model. As shown in Figure 7, by combining both approaches, we can observe that the feature importance ranking of the RF model is as follows: CRP, ALB, PCT, Progressive dyspnea, and Hb. Among these, CRP, PCT, and Progressive dyspnea showed a positive correlation with the occurrence of SPCP. Higher values of CRP and PCT were associated with an increased likelihood of progressing to SPCP. On the other hand, ALB and Hb demonstrated a negative correlation with SPCP, indicating that lower values were associated with a higher risk of developing SPCP.

## 4. Discussion

In this investigation, the prevalence of SPCP was found to be 20.5%. A notable distinction in mortality rates during hospitalization was observed between SPCP and NSPCP groups, with a significantly higher mortality rate in the SPCP group (61.1% vs. 0%, *p* < 0.05). Remarkably, kidney transplant patients with SPCP exhibited a significantly worse prognosis. Therefore, timely identification and intervention in individuals at high risk are of paramount importance. To evaluate the prognosis of 88 patients who experienced PCP following kidney transplantation, this research employed multiple machine learning algorithms.

Among the variables analyzed in this study, CRP emerged as the most prominent. CRP is an acute-phase protein synthesized by the liver in response to inflammation and is commonly utilized as an indicator of systemic inflammation and infection [12,13]. CRP rises rapidly during the initial stages of infection, with elevated values correlating positively with the severity of the infection or inflammation, and can serve as a sensitive biomarker for many inflammatory diseases, such as infections and tissue damage [14,15]. And it can serve as one of the predictors of severe COVID-19 [16]. PCP is caused by the opportunistic pathogen *Pneumocystis carinii*. Sun et al. assessed blood biomarkers in 32 HIV-infected PCP patients and found that CRP levels were significantly higher in patients who were critically ill or died [17]. Elevated CRP has also been previously reported to be strongly associated with disease severity and death in HIV-infected PCP patients [18]. A multicenter study by Shiba et al. on PCP patients with rheumatoid arthritis similarly found that CRP levels at baseline were significantly higher in the PCP mortality group than in the survivor group [19]. Hou et al. found that advanced age, high baseline levels of LDH and CRP, and low platelet counts were risk factors for in-hospital mortality in patients with mNGS-positive PCP diagnosed by second-generation sequencing of macrogenomi [20]. The numerous evidence above indicates the important role of CRP in assessing the severity of PCP patients, and the possible mechanism is that when the immune system detects the presence of *Pneumocystis carinii*, it triggers an inflammatory response, which in turn activates immune cells to release inflammatory cytokines. And CRP can play a crucial role in the immune response against infection by participating in the activation of the complementary system [12,21]. Consequently, CRP levels can serve as a potential biomarker for evaluating the progression and severity of PCP in kidney transplant recipients.

ALB, a significant protein synthesized by the liver, plays crucial roles in maintaining osmolarity and facilitating substance transport in the bloodstream. ALB reduces inflammatory exudation in the lungs and also has antioxidant properties, and serum ALB levels can decrease during oxidative stress or infection, and its antioxidant properties can be impaired, ultimately leading to further tissue damage [22]. The existing literature has demonstrated a correlation between decreased ALB levels and severe pneumonia [23]. Serum ALB levels were considered to be good predictive markers for morbidity and mortality of critically ill patients [24,25,26]. Previous studies have also shown that low ALB levels significantly increase pneumonia mortality in kidney transplant recipients [27,28]. Because of the association between low serum albumin levels and increased disease severity, serum albumin could theoretically also be used as a predictive biomarker of disease severity in patients with PCP. Therefore, close monitoring of ALB levels and appropriate supplementation may aid in slowing the progression of PCP.

According to the SHAP plot, the third-ranked variable in the RF model is PCT, a precursor protein for calcitonin. PCT is primarily synthesized by the C-cells of the thyroid gland. It is worth noting that PCT is predominantly released into the bloodstream as a response to bacterial infection and systemic inflammation. During infection, PCT plays a crucial regulatory role in modulating the immune response. It exerts inhibitory effects on the production of pro-inflammatory cytokines, such as TNF-α and IL-6, while promoting the release of anti-inflammatory cytokines, such as IL-10. This regulatory function is vital for controlling the inflammatory response and minimizing tissue damage. PCT is also frequently used in the intensive care unit as an indication for infection, severity of illness and antibiotic therapy [29,30,31]. Despite Pneumocystis not being a bacterium, its presence still triggers an immune response that can result in elevated PCT levels. In the case of PCP, higher PCT levels can serve as an indicator of the intensity of the immune response and the severity of the infection. Elevated PCT levels are associated with more SPCP and unfavorable clinical outcomes. Monitoring PCT levels can assist in assessing disease progression and guiding the treatment of kidney transplant recipients affected by PCP.

The occurrence of progressive dyspnea upon admission to hospital often indicates a more severe form of PCP and suggests that the patient may have missed out on early intervention and treatment. Previous studies have shown that early intervention and treatment are critical to the prognosis of non-HIV patients with PCP [32,33,34]. In addition, non-HIV-infected patients with PCP have faster disease progression, poorer outcomes, higher mortality rates, and a higher risk of co-infections compared to HIV-infected patients, with major symptoms typically including exertional dyspnea, dry cough, and fever [33,35,36,37,38]. The presence of progressive dyspnea, on the other hand, often indicates the presence of a persistent inflammatory process in the lungs with concomitant damage to the alveolar–capillary membrane, and as the infection progresses, respiratory muscle fatigue or even respiratory failure may occur [32]. In summary, the evaluation of progressive dyspnea upon admission holds great clinical importance as it enables the prevention of further respiratory damage and contributes to the improvement of the patient’s prognosis.

PCP has the potential to impede efficient exchange of oxygen and carbon dioxide in the lungs. Hb, the protein found in red blood cells responsible for oxygen transport, plays a critical role in maintaining tissue oxygenation. When Hb levels decrease, the capacity for oxygen transportation is reduced. This impairment can disrupt the body’s ability to deliver sufficient oxygen to tissues and organs, including the lungs themselves. Previous studies have found that severe COVID-19 is associated with lower red blood cell counts in patients, and this may also apply to Hb [39]. Walzer et al. [40] and Miller et al. [41] showed that low Hb levels on admission were an independent risk factor for death in HIV-infected PCP patients. Therefore, the detection of Hb in admitted patients may be clinically important for the prognostic assessment of patients with PCP after renal transplantation.

In the realm of PCP, various prior studies have been conducted to predict the risk of ICU admission and mortality in affected patients. One study by Benjamin et al. examined 107 immunocompromised PCP patients, including 25% with solid-organ transplantation. Their findings revealed that older age and the presence of P. jirovecii oocysts in bronchoalveolar lavage (BAL) examination were independently associated with SPCP [42]. Fan et al. utilized nomogram to predict the risk of death and ICU admission in non-HIV-infected PCP patients. Their research identified respiratory rate, dyspnea, lung moist rales, LDH, BUN, CRP/ALB ratio, and pleural effusion as potential predictors for ICU admission risk [43].

In contrast to previous studies, our research employed six machine learning models for prediction. Through parameter tuning and validation, we identified the RF model as the best-performing model in the validation set, exhibiting an AUC value of 0.92. Notably, the variables utilized in this model consisted of laboratory test results and clinical symptoms, which are easily obtainable and allow clinicians to determine the likelihood of SPCP development upon hospital admission for patients with PCP following kidney transplantation.

However, it is important to acknowledge some limitations of this study. Firstly, it is a single-center observational study, making it susceptible to potential biases. Secondly, despite utilizing SMOTE to address imbalanced data, it is crucial to recognize that the sample size in this study was limited. To further enhance the accuracy of the model, we plan to gather additional clinical data and optimize the parameters in future research.

## 5. Conclusions

We used multiple machine learning approaches for predicting the risk of SPCP in patients with PCP after kidney transplantation. Machine learning allows for early and accurate estimation of SPCP risk in PCP patients at the time of admission, which may provide guidance for clinical decision making.

## Figures and Tables

**Figure 1 diagnostics-13-02735-f001:**
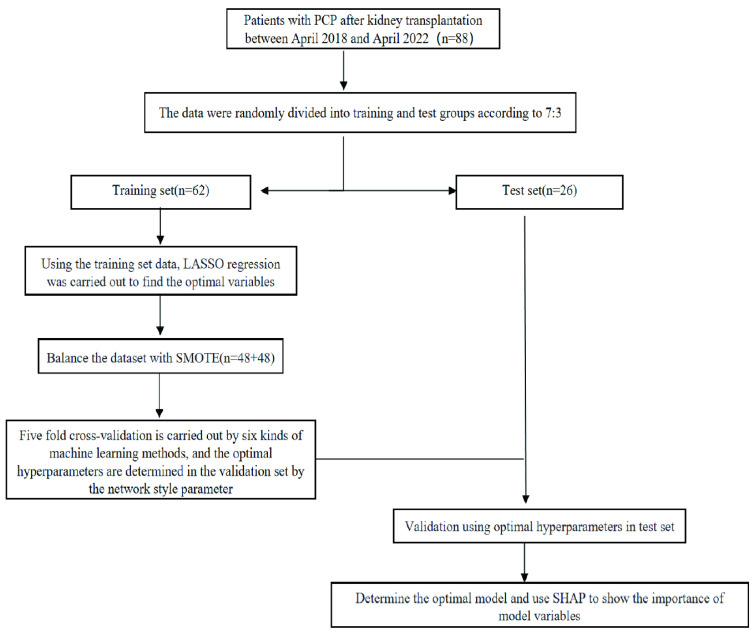
Flowchart of the process of this study.

**Figure 2 diagnostics-13-02735-f002:**
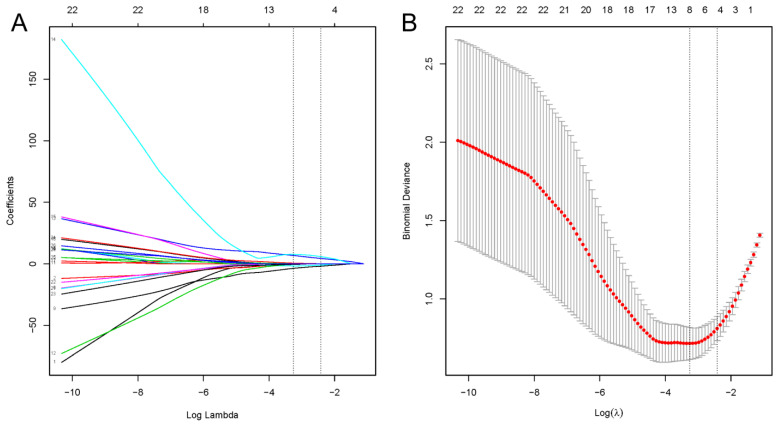
Least absolute shrinkage and selection operator (LASSO) regression to identify clinical features that may predict kidney transplant patients with severe PCP (SPCP). (**A**) LASSO coefficient profiles. (**B**) LASSO regression using 5-fold cross-validation and the “minimum plus one standard error” criterion to identify the optimal penalization coefficient lambda (λ).

**Figure 3 diagnostics-13-02735-f003:**
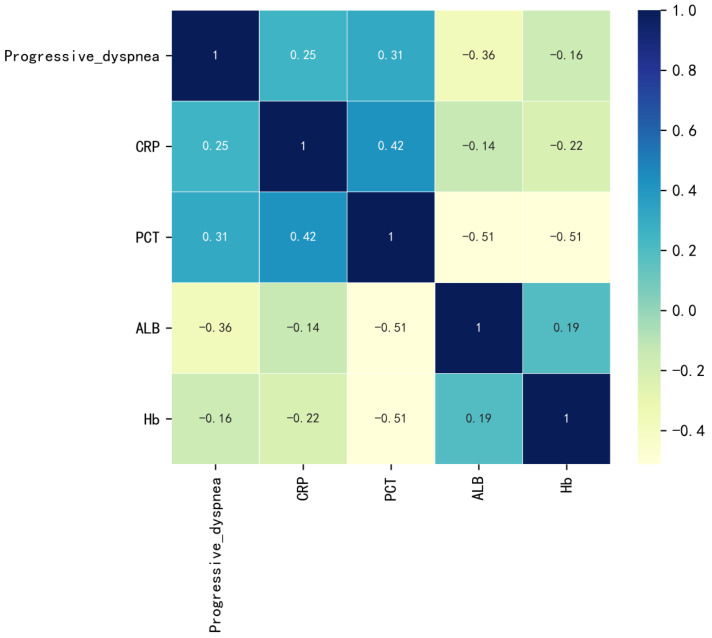
Heatmap of correlations among the five variables. CRP, C-reactive protein; PCT, procalcitonin; ALB, albumin; and Hb, Hemoglobin.

**Figure 4 diagnostics-13-02735-f004:**
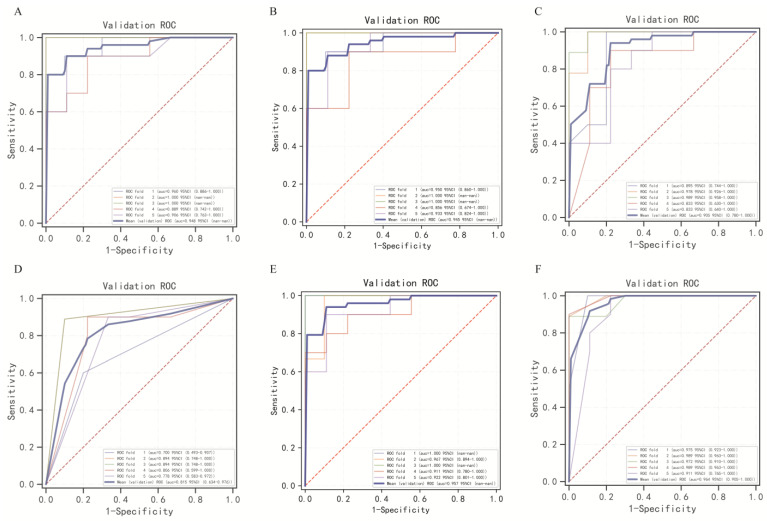
ROC curves for internal validation sets of six machine learning algorithms. (**A**) Random Forest model; (**B**) Logistic Regression model; (**C**) eXtreme Gradient Boosting model; (**D**) Light Gradient Boosting Machine model; (**E**) Support Vector Machine model; and (**F**) K-Nearest Neighbor model.

**Figure 5 diagnostics-13-02735-f005:**
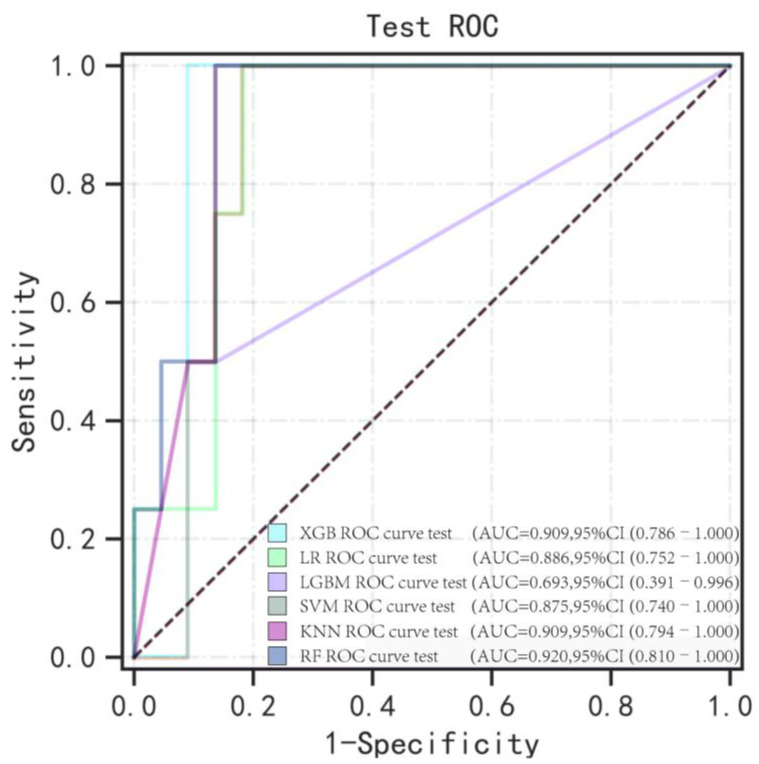
ROC curves for the test set of six machine learning algorithms. RF, Random Forest; LR, Logistic Regression; XGB, eXtreme Gradient Boosting; LGBM, Light Gradient Boosting Machine; SVM, Support Vector Machine; and KNN, K-Nearest Neighbor.

**Figure 6 diagnostics-13-02735-f006:**
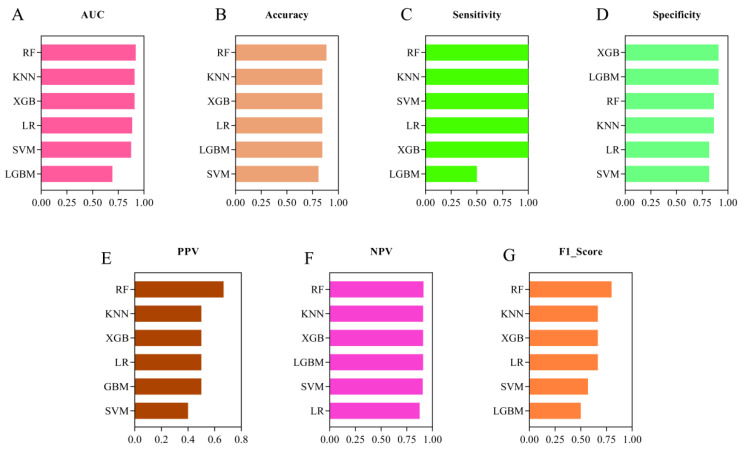
Visualization of the specific performance of the six machine learning algorithms in the test set. RF: Random Forest; LR: Logistic Regression; XGB: eXtreme Gradient Boosting; SVM: Support Vector Machine; KNN: K-Nearest Neighbor; LGBM: Light Gradient Boosting Machine; AUC: area under the curve; PPV: positive predictive Value; and NPV: negative predictive value.

**Figure 7 diagnostics-13-02735-f007:**
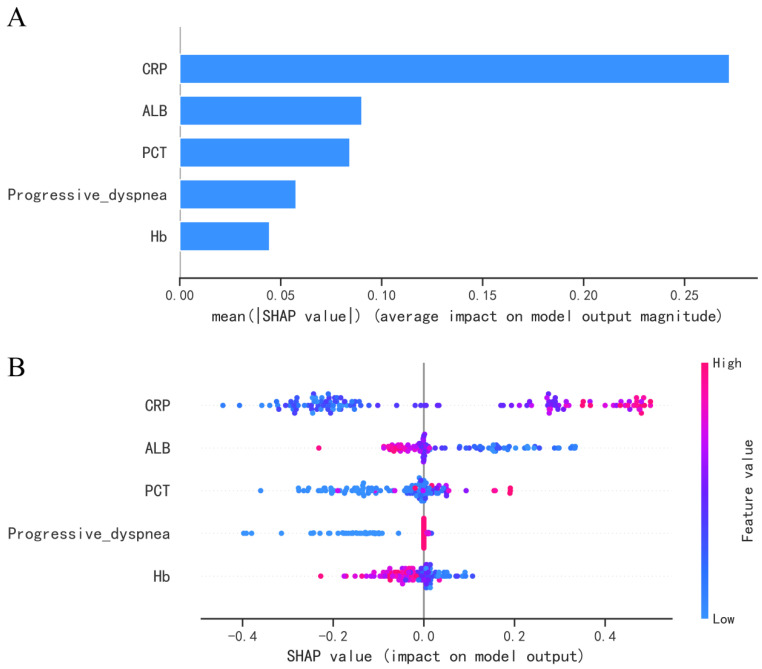
Interpretation of models using Shapley Additive Explanations (SHAP) analysis. (**A**) Relative importance of variables based on SHAP for random forest (RF) prediction model. (**B**) SHAP summary plot of RF model with LASSO selection features. CRP, C-reactive protein; PCT, procalcitonin; ALB, albumin; Hb, Hemoglobin.

**Table 1 diagnostics-13-02735-t001:** Baseline demographic and clinical laboratory examination characteristics of all patients (training and test groups).

	All, *n* = 88	Training, *n* = 62	Test, *n* = 26	*p*-Value
Age, yr	43.97 ± 9.93	42.81 ± 10.22	46.73 ± 8.78	0.091
Male, *n*	60 (68.2%)	37 (59.7%)	23 (88.5%)	0.017
Post-transplant time, months	7.00 [5.00, 10.00]	7.00 [5.00, 10.00]	7.00 [4.25, 10.00]	0.72
Clinical presentation				
Fever, *n*	64 (72.7%)	45 (72.6%)	19 (73.1%)	1
Dry cough, *n*	25 (28.4%)	17 (27.4%)	8 (30.8%)	0.953
Progressive dyspnea, *n*	49 (55.7%)	38 (61.3%)	11 (42.3%)	0.161
Laboratory results				
GM test number	469.30 [174.74, 600.00]	436.35 [162.26, 600.00]	571.00 [253.41, 612.15]	0.423
GM test, *n*	76 (86.4%)	53 (85.5%)	23 (88.5%)	0.975
CRP, mg/L	71.10 [43.70, 92.75]	76.00 [44.18, 99.17]	61.55 [41.75, 75.62]	0.101
PCT, ng/ml	0.20 [0.10, 0.34]	0.20 [0.11, 0.40]	0.18 [0.10, 0.30]	0.711
PCT ≥ 0.5 ng/mL, *n*	17 (19.3%)	14 (22.6%)	3 (11.5%)	0.367
Neut, 1 × 10^9^	6.26 [4.16, 8.81]	6.10 [4.41, 8.81]	6.60 [3.83, 8.57]	0.671
Hb, g/L	113.00 [98.50, 121.25]	113.00 [96.00, 121.00]	113.50 [102.25, 123.50]	0.355
Plt, 1 × 10^9^	210.00 [163.50, 246.50]	212.00 [175.75, 239.50]	188.50 [150.00, 247.50]	0.209
BUN, mg/dL	12.28 [8.96, 19.16]	12.60 [9.21, 19.38]	12.00 [8.77, 17.62]	0.791
Glu, mmol/L	6.35 [5.30, 8.20]	6.12 [5.26, 7.80]	7.12 [5.75, 8.39]	0.167
WBC, 1 × 10^9^	7.69 [5.02, 10.14]	7.69 [5.36, 10.27]	7.58 [4.76, 9.85]	0.586
LYM, 1 × 10^9^	0.53 [0.36, 0.89]	0.52 [0.34, 0.86]	0.53 [0.40, 0.90]	0.742
ALB, g/L	37.40 [33.65, 40.00]	37.75 [34.35, 40.00]	35.05 [33.05, 39.30]	0.583
GLB, g/L	21.25 [18.80, 23.79]	21.15 [18.65, 23.38]	21.80 [19.30, 24.09]	0.22
Scr, μmol/L	151.50 [122.50, 195.25]	158.00 [134.25, 198.25]	134.50 [113.50, 172.00]	0.034
Previous history				
ATG induction, *n*	33 (37.5%)	23 (37.1%)	10 (38.5%)	1
ATG dose, mg	0.00 [0.00, 75.00]	0.00 [0.00, 75.00]	0.00 [0.00, 75.00]	1
Rejection, *n*	39 (44.3%)	29 (46.8%)	10 (38.5%)	0.63
CMV infection, *n*	40 (45.5%)	31 (50.0%)	9 (34.6%)	0.031
ICU, *n*	18 (20.5%)	14 (22.6%)	4 (15.4%)	0.636
Death, *n*	11 (12.5%)	8 (12.9%)	3 (11.5%)	1
Duration of hospitalization, days	28.50 [19.00, 36.25]	29.00 [19.25, 37.00]	28.00 [17.25, 36.00]	0.277

GM test: glactomannan test; CRP: C-reactive protein; Scr: serum creatinine; PCT: procalcitonin; WBC: white blood cell; LYM: lymphocyte; Neut: Neutrocyte; Hb: Hemoglobin; Plt: platelet count; BUN: Blood Urea Nitrogen; Glu: Glucose; ALB: albumin; GLB: globulin; ATG: antihuman thymocyte globulin; PJP: pneumocystis jiroveci pneumonia; CMV: cytomegalovirus; and PSI: pneumonia severity index.

**Table 2 diagnostics-13-02735-t002:** Summary of the specific performance of the six machine algorithm models in the test set.

	AUC	Accuracy	Sensitivity	Specificity	PPV	NPV	F1_Score
RF	0.920 (0.810–1.000)	0.885	1	0.864	0.667	0.913	0.8
LR	0.886 (0.752–1.000)	0.846	1	0.818	0.5	0.875	0.667
XGB	0.909 (0.786–1.000)	0.846	1	0.909	0.5	0.909	0.667
SVM	0.875 (0.740–1.000)	0.808	1	0.818	0.4	0.905	0.571
KNN	0.909 (0.794–1.000)	0.846	1	0.864	0.5	0.909	0.667
LGBM	0.693 (0.391–0.996)	0.846	0.5	0.909	0.5	0.909	0.5

RF: Random Forest; LR: Logistic Regression; XGB: eXtreme Gradient Boosting; SVM: Support Vector Machine; KNN: K-Nearest Neighbor; LGBM: Light Gradient Boosting Machine; AUC: area under the curve; PPV: positive predictive Value; and NPV: negative predictive value.

## Data Availability

Data are available upon request due to privacy/ethical restrictions.

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
