# Peer review of "Machine Learning Models for Prediction of Severe Pneumocystis carinii Pneumonia after Kidney Transplantation: A Single-Center Retrospective Study"

_diagnostics, 2023, doi:10.3390/diagnostics13172735_

Round 1

Reviewer 1 Report

1. Add more keywords; and make them different from the title, for example; artificial intelligance.

2. No need for the word "infection" after PCP.

3. Line 37-43: Although you cite a slightly relevant and recent article, not all of the information you mentioned are from the cited reference. Consider the following recent review:

Alsayed, A. R., Al-Dulaimi, A., Alkhatib, M., Al Maqbali, M., Al-Najjar, M. A. A., & Al-Rshaidat, M. M. (2022). A comprehensive clinical guide for Pneumocystis jirovecii pneumonia: A missing therapeutic target in HIV-uninfected patients. Expert Review of Respiratory Medicine, 16(11-12), 1167-1190.    

4. Line 45: what is SPCP?

5. At the end of introduction you mentioned few studies about machine learning and PCP; also, consider the following recent paper:

Lang, Q., Li, L., Zhang, Y., He, X., Liu, Y., Liu, Z., & Yan, H. (2023). Development and Validation of a Diagnostic Nomogram for Pneumocystis jirovecii Pneumonia in Non-HIV-Infected Pneumonia Patients Undergoing Oral Glucocorticoid Treatment. Infection and Drug Resistance, 755-767.  

6. Line 75: the included patients is more suitable in the results rather than the methods (already mentioned in Line 149).

7. Line 243-249: needs a reference.

8. Line 251-260: lack of cited references.

9. In general, the authors performed little literature review. The discussion section should be improved. Although the authors cited some references about predictors of COVID-19 severity when they mentioned the lab parameters, however, more recent articles are available.

Few errors found

Author Response

Dear reviewer ,

we have revised our manuscript one by one according to your professional review opinions, reintroduced references, and conducted a new literature review in the discussion section. The modified part is shown in red font.

  1. It has been added
  2. It has been deleted
  3. It has been added
  4. Added remarks: severe Pneumocystis jirovecii pneumonia (SPCP)
  5. It has been added
  6. It has been resolved
  7. It has been resolved
  1. It has been resolved
  2. It has been resolved

Thanks again for the reviewer's professional comments

Yiting Liu

Reviewer 2 Report

Summary of the Manuscript:

Reviewed manuscript titled "Machine learning models for prediction of Severe Pneumocystis jirovecii Pneumonia after kidney transplantation: a single-center retrospective study" by Yiting Liu al. The article presents a retrospective study aiming to develop and validate a prognostic model for predicting postoperative severe Pneumocystis jirovecii pneumonia (SPCP) in kidney transplant recipients using machine learning algorithms. The study was conducted at the Department of Organ Transplantation, Renmin Hospital of Wuhan University, and involved 88 patients who underwent kidney transplantation.

Strengths

·         Data Collection: The authors have meticulously gathered a broad spectrum of variables, from demographic data to laboratory results. This comprehensive data collection is commendable and provides a holistic view of the patient's condition.

·         Methodological Rigor: The use of LASSO regression for variable screening and Spearman correlation analysis for multicollinearity indicates a statistically sound approach to data analysis.

·         Model Diversity: The study's decision to employ multiple machine learning models is a strength, allowing for a robust comparison of predictive capabilities.

·         Interpretability: The application of the SHAP algorithm is a significant step towards model transparency and interpretability in the realm of machine learning.

Weaknesses

  • Sample Size: The study's reliance on 88 patients (Lines 67-75) might be a limitation. For machine learning models, especially with the depth of variables considered, a larger sample size could enhance the model's robustness.
  • Single-center Analysis: The study's confinement to a single-center (Lines 6-10) might limit the generalizability of its findings. A multicenter approach could provide a more diverse dataset.
  • Outcome Measures: While the primary outcome of interest is the occurrence of SPCP during hospitalization (Lines 76-79), it would be beneficial to have more granular outcome measures, especially given the depth of the input variables.

Recommendations

  • Dataset Expansion: Given the depth of the variables considered (Lines 85-95), it would be statistically prudent to increase the sample size or collaborate with other centers.
  • Prospective Validation: A prospective validation on a new cohort of patients (outside the retrospective dataset) would provide a more robust evaluation of the model's real-world applicability.
  • Feature Engineering: Given the study's focus on machine learning, there might be an opportunity to delve deeper into feature engineering (Lines 96-108). Combining certain clinical and laboratory parameters or creating interaction terms might enhance the model's predictive power.

Conclusion

The article provides a statistically rigorous approach to a clinically relevant problem. The depth of data collection, methodological rigor, and focus on model interpretability are commendable. However, there are areas, especially related to dataset size and multicenter validation, where the study could be enhanced.

Author Response

Dear reviewer,

Thank you very much for the reviewer's professional advice and comprehensive analysis of our article. As for the problem of sample size, due to the relatively small sample size of PCP after kidney transplantation, the sample size of our paper is relatively larger than those published in the past. In addition, our endpoint outcome event was severe PCP, we gave SMOTE treatment in the training set, so that the sample size of severe PCP could be expanded in a scientific way to alleviate the defect of insufficient sample. In response to the problem of single-center data sets, we acknowledge its shortcomings and have pointed out in the limitations. In terms of outcome measurement, only SPCP was measured, but SPCP is closely related to mortality and is a very important outcome indicator with strong clinical significance. Thanks again for the reviewer's professional advice. If the article is published, we will lead a multi-center research based on this article in the future and gradually improve this research.

Thanks again for the reviewer's professional comments

Yiting Liu

Reviewer 3 Report

The manuscript seems to be well-written. I have some minor comments.

1.         (Methods) It seems that analysis methods for binary data were used in the analysis. However, it is better to conduct survival analysis rather than binary data analysis. Why didn’t the authors conduct survival analysis?

2.         (Methods, Hyperparameter optimization) It is written that six machine learning methods were selected for hyperparameter optimization. However, I think that six machine learning methods were selected for the prediction purpose. Is the sentence correct?

3.         (Methods) How did the authors deal with missing data?

4.         (Results) I think that the aim of this study is to develop a prediction model for PCP. However, it is difficult to use the developed RF model for others. Will the model become available to the public? Or, it might be better to show an equation of prediction model by a logistic regression model.

It seems to be OK.

Author Response

Dear reviewer,

  1. Thank you for your valuable advice. We previously considered using survival analysis, but the time span from admission to diagnosis of severe PCP is relatively short, and the time variable is negligible when showing how long patients stay in hospital, so we did not use survival analysis.
  2. Thanks for your valuable advice, these six kinds of machine learning have different functions on the training set and the verification set. In the training set, six kinds of machine learning hyperparameters are optimized, and the best hyperparameters of each machine learning are determined. In the validation set, each machine learning method uses the best hyperparameters, and six machine learning methods are compared to determine which machine learning method performs best.
  3. KNN method was used to interpolate the missing data in the previous stage, and we made corresponding supplements in the paper and marked them with red font
  4. At present, multiple machine learning methods have not been used to predict the severity rate of PCP patients after kidney transplantation, which significantly increases the risk of death; We compared the results of multiple machine learning evaluations and ultimately found that the random forest model outperformed traditional logistic regression on the test set. Using the previous data, our research group has written an article to evaluate the prognosis by simple logistic regression, which is still under submission. In the last article, we presented a generalized prediction model. This paper focuses on comparing the predictive performance of various machine learning methods, identifying random forests as the most important method, and performing interpretable SHAPs to visualize each variable. We all know that the predictive performance of machine learning methods is high, but because the black box principle is difficult to explain, it is impossible to create a simple nomogram to show. In addition, this study is a preliminary exploration. After the release of the results of this study, we will actively cooperate with the multi-center research and eventually cooperate with professional and technical personnel to conduct predictive display of machine learning, but it may not be possible at this stage.Thanks again for the reviewer's professional comments.

Yiting Liu

Round 2

Reviewer 1 Report

The manuscript is better now.

Good